# Non-Invasive SWIR Monitoring of White Marble Surface of the Cathedral of Santa Maria del Fiore (Florence, Italy)

Silvia Vettori [1,*], Davide Romoli [2], Teresa Salvatici [2], Valentina Rimondi [2], Elena Pecchioni [2], Sandro Moretti [2], Marco Benvenuti [2,3], Pilario Costagliola [2], Rachele Manganelli Del Fà [1], Michele Coppola [4], Beatrice Agostini [5] and Francesco Di Benedetto [6,*]

1   Institute of Heritage Science, National Research Council (ISPC-CNR), Sesto Fiorentino, 50019 Florence, Italy
2   Department of Earth Sciences, University of Florence, 50121 Florence, Italy
3   Institute of Geosciences and Earth Resources, National Research Council (IGG-CNR), 50121 Florence, Italy
4   Department of Architecture, University of Florence, 50121 Florence, Italy
5   Opera del Duomo di Firenze, 50121 Florence, Italy
6   Department of Physics and Earth Sciences, University of Ferrara, 44122 Ferrara, Italy
*   Correspondence: silvia.vettori@cnr.it (S.V.); francesco.dibenedetto@unife.it (F.D.B.)

**Abstract:** The monitoring of stone alteration represents a key factor in the knowledge and prediction of the status of conservation of building stones in the urban framework. A continuous monitoring requires a non-destructive analytical approach and, possibly, a simple, low-cost and effective tool to study the decay processes. Previous studies demonstrated the capability of the SWIR hyperspectral technique to gain information on the degree of sulfation of carbonate stone surfaces. In this study we aim at setting up a protocol to investigate *on-site* the sulfation degree of the white marble cladding surfaces of the worldwide-famous Santa Maria del Fiore Cathedral in Florence (Italy). The proposed protocol couples information by SWIR hyperspectral and colorimetric techniques. We have proved that, in selected areas investigated at a distance of nine years, the colour and the mineralogical changes (i.e., sulfation) are significantly greater than the relative uncertainties of the two methods. Moreover, the proposed protocol results rapid, repeatable and fully not invasive.

**Keywords:** Santa Maria del Fiore Cathedral; SWIR monitoring; white marble; gypsum; calcite

## 1. Introduction

The continuous monitoring of the conservation state of outdoor natural (stone) surfaces, including the characterization and quantification of the neo-formation and deposition of materials, is considered a good practice for timely planning conservative interventions and therefore for preserving historical buildings [1–4].

All geomaterials, exposed as a natural outcrop or in a building, are subject to the physical, chemical and biological weathering. The decay processes of stone materials depend on several factors, both intrinsic, such as the mineralogical and textural characteristics of rocks, and extrinsic, such as the environmental conditions (climate and microclimate affecting the whole building or part of it, superimposed by human activities). Atmospheric pollution, frost and salt weathering are traditionally considered the major causes of building stone decay [5,6]. Main damages include surface corrosion, soiling and biodegradation, with consequent loss of details, blackening and formation of crusts on stone surfaces. In recent years, major changes occurred in both the sources and amounts of air pollutants emissions that have deeply modified the rate and the extent of historical buildings damage. One of the most diffused degradation processes affecting both natural and artificial carbonate materials, specifically marble exposed to the urban atmosphere, is the sulfation reaction, i.e., the formation of sulphate-based deposits (i.e., "black crusts") [7]. Black crusts are typically composed by gypsum, replacing the primary calcite, embedding different types of particles [8–16]. The volume increase due to the formation of the gypsum from calcite

is the major cause of the development of cracks in the carbonate substratum of black crusts, ultimately leading to degradation of stone materials. Besides loss of shape, sulfation reaction may cause a colour change to the material. Among the processes able to act on the process of carbonate sulfation, one can cite the changes in vehicular traffic and the climate changes. The decreased content of acidic pollutants in urban atmosphere during the late 20th century is held responsible for the observed decreasing rate of formation of degradation crusts in the last years: more precisely, although $SO_x$ and $NO_x$ are thought to have decreased in the recent past due to tighter environmental regulations or higher public expenditure to improve air quality in high-income European areas [17], at present, sulfation is still among the most relevant alteration processes of carbonate stone buildings in the urban environment [18,19]. Conversely, there is a common consensus, indeed, that new climate regimes will cause dramatic changes in blackening patterns due to the probable future increases in atmospheric $CO_2$ [20].

Accordingly, the monitoring of stone alteration represents a key factor in the knowledge and prediction of the trend affecting the building stone in the 21th century urban framework [21,22]. Continuous monitoring requires a non-destructive analytical approach and, possibly, a simple, low-cost and effective tool to study the decay processes affecting the architectural heritage and artworks in general.

The coexistence of gypsum and calcite is relatively easy to be investigated by hyperspectral techniques, because of the mineral characteristic absorption overtone bands in the Short-Wave InfraRed (SWIR) range [23–26]. Hyperspectral techniques are methods able to couple lateral resolution with energy/wavelength resolution, so as to obtain either compositional information on a region or spectral information on a point, depending on the needs. In particular, the SWIR range is useful to highlight spectral contribution due to the intensity of bands arose from vibrational transition linked to specific mineralogical phases. Previous studies on stone alteration demonstrated the following:

- The possibility to attain information on the degree of sulfation of carbonate stone surfaces through the set-up of artificial alteration experiments of carbonate stone specimens under laboratory conditions [27];
- The capability of SWIR hyperspectral technique to on-site monitor the most effective cleaning procedures for black crusts on marble surfaces [28].

In this study, we aim at setting up a protocol for the on-site investigation of the sulfation degree of the white marble cladding surfaces of the worldwide famous Santa Maria del Fiore Cathedral in Florence (Italy) (Figure 1). A non-invasive, reliable and extremely rapid diagnostic protocol is thus provided to monitor the conservation state of outdoor surfaces, including the characterization and quantification of the neo-formation and deposition of materials over time. The proposed protocol couples information by SWIR hyperspectral and colorimetric techniques.

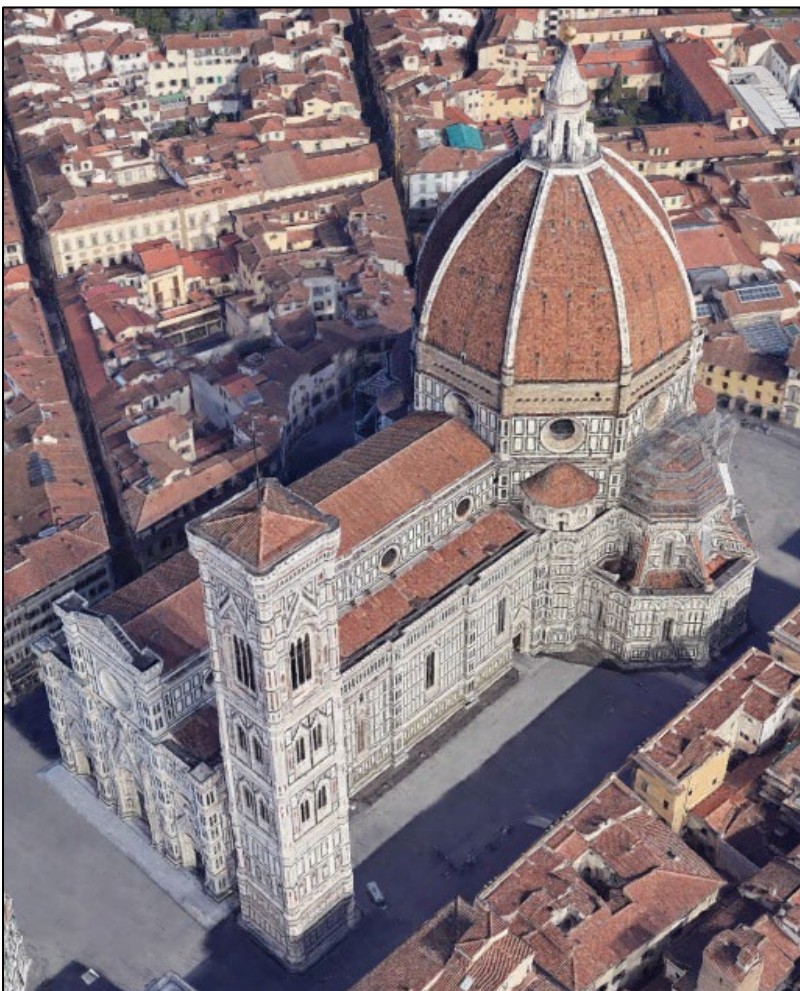

**Figure 1.** Aerial photo of Santa Maria del Fiore Cathedral in Florence (by Google Earth).

## 2. Materials and Methods

### 2.1. Site Description

The on-site monitoring was carried out on the white marble cladding of the Santa Maria del Fiore Cathedral, a church that plays an outstanding role in the historical architecture of the city. The construction of the Cathedral started in the late 13th century to replace the small church of Santa Reparata, which stood in front of San Giovanni Baptistery. From the laying of the first stone on the 8 September 1296 following the Arnolfo di Cambio project until the structural completion of the Filippo Brunelleschi dome in 1436, several famous artists (such as Giotto, Francesco Talenti, Andrea Pisano) were involved in its construction, leaving enduring marks of their architectural expressions. The exterior cladding decoration was obtained through a trichromy by using white marble, green serpentinite and red limestones [29]. The façade, planned by Emilio De Fabris, is a later integration, built in the 19th century; it was inaugurated in 1887 [30–32].

The white marble of the original construction of the cathedral was quarried in the Apuan Alps [29–33], one of the best-known stone materials (Carrara Marbles), used by architects and artists throughout the world. The Santa Maria del Fiore Cathedral white marble exhibits various forms of decay, such as deposits, crusts, erosion, mechanical damage and biological colonization (patinas and discoloration).

### 2.2. Acquisition Data Procedure

In this study, we examined one of the most relevant decay processes affecting both natural and artificial carbonate materials exposed to the urban atmosphere: the formation

of sulphate-based deposits. The methodology consists of the semiquantitative detection of gypsum on the carbonate stone surfaces, which is considered the precursor symptom of damage as a consequence of the reaction between carbonate matrices (i.e., marble) and atmospheric pollution ($SO_x$). Moreover, a colorimetric investigation of the same surface was performed because it could allow us to link changes in the mineralogical composition and changes in the perceived colour of the stone cladding.

To evaluate the trend of the stone degradation process (i.e., sulfation), two acquisition data campaigns were repeated nine years apart (2012 and 2021) on the same representative points (n = 24) of the cathedral.

The studied points were in correspondence of four different areas of the cathedral (Figures 2 and 3):

- from 1B to 6B in front of Via de' Servi street on the north side of the building;
- from 7B to 10B on the Apse on the south-east side;
- from 11B to 18B on the Porta dei Canonici on the south side;
- from 19B to 24B on the north side.

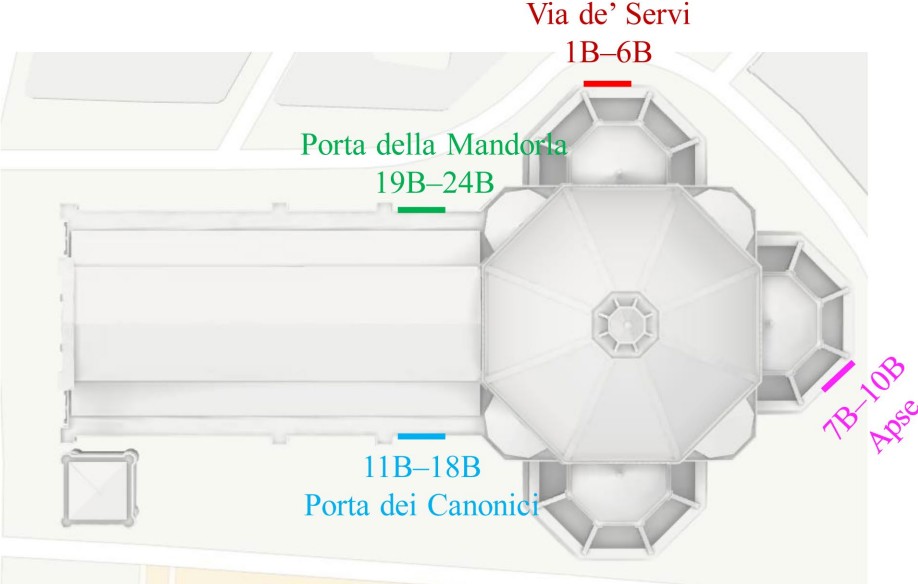

**Figure 2.** Plan of the Santa Maria del Fiore Cathedral. The location of the investigated spot areas is reported. The region highlighted in light pink traces the cladding object of an extraordinary cleaning, as reported in the text (Map modified from OpenStreetMap).

To better locate the studied points with time, a marked polyacetate mask, provided by holes with a diameter equal to the spot sizes of the instrumentations (Figure 4a–c), was used, together with an accurate photographic documentation and macroscopic description. The areas were selected based on the different state of decay of the stone material (in 2012) but also depending on the different exposure to atmospheric agents and orientations.

In these nine years (2012–2021), only the Apse side was subject to extraordinary cleaning interventions in 2013. The cleaning operation was performed with sepiolite and Arbocel poultices with 10% of ammonium carbonate. Where biological growth was active, a spray or brush commercial biocide treatment was also performed. All treated surfaces were finally cleaned with deionized water and natural sea sponges.

The on-site monitoring was performed using:

- A portable spectrophotometer to obtain the colorimetric coordinates (Figure 4b);
- A field portable high-resolution spectroradiometer to collect reflectance measurements (Figure 4a).

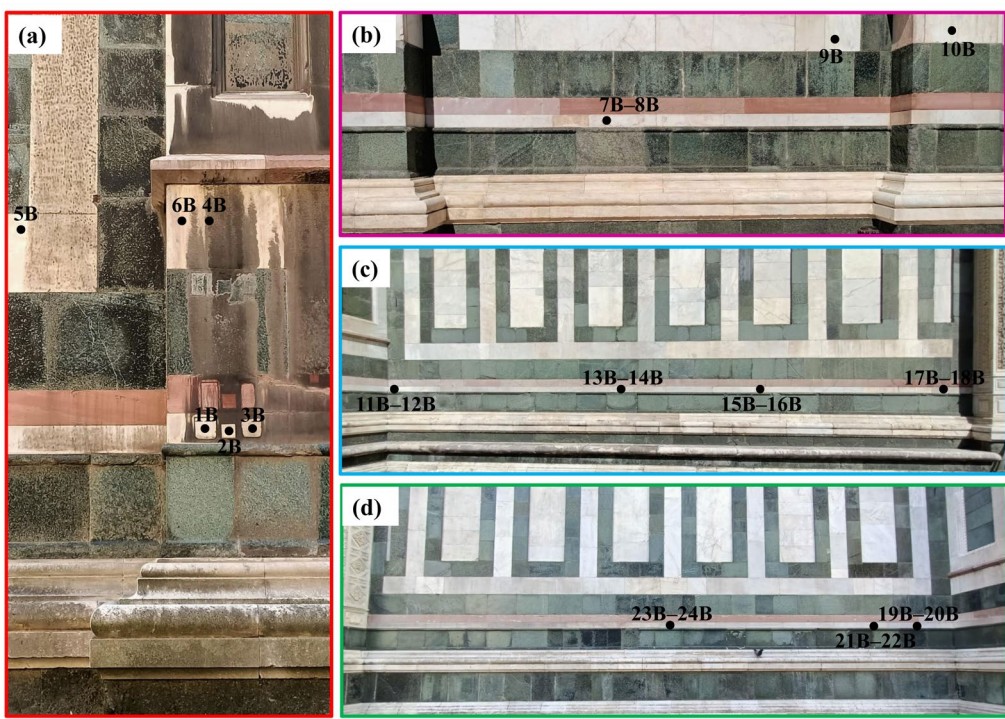

**Figure 3.** The location of the twenty-four investigated spot areas: (**a**) Via de' Servi, north side; (**b**) Apse, south-east side; (**c**) Porta dei Canonici, south side; (**d**) Porta della Mandorla, north side. Photographs taken by the authors.

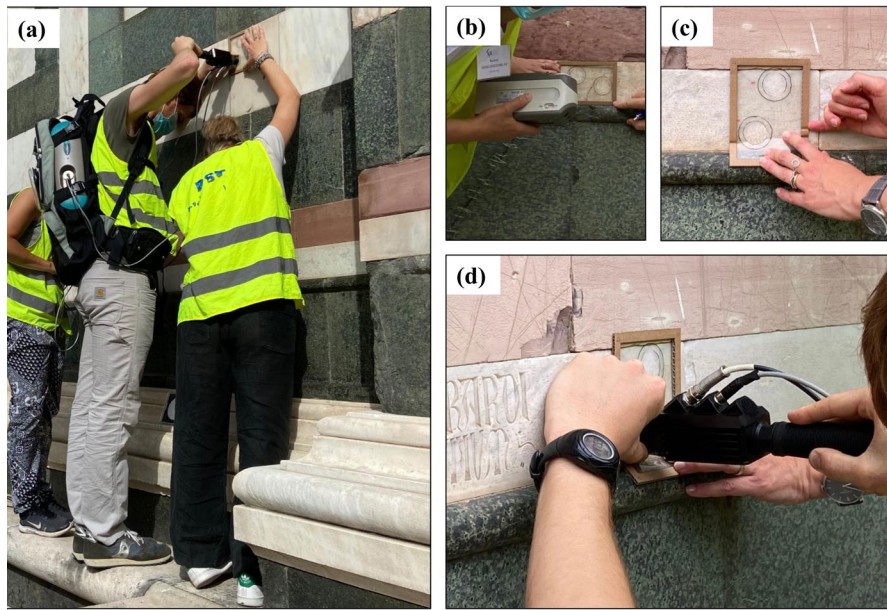

**Figure 4.** Some moments during the measurement campaign: (**a**,**b**) spectroscopic and colorimetric investigations; (**c**) detail of the polyacetate mask; (**d**) detail of the contact reflectance probe. Photographs taken by the authors.

## 2.3. Colorimetric and Spectroscopic Investigations

Colorimetric analyses were performed using a CM-2600d Konica-Minolta spectrophotometer equipped with an integrative sphere inside the apparatus and a Xenon lamp to pulse the light on the sample surface (Figure 4d), with a measurement aperture of 8 mm and light reflection from the surface at an angle of 8°. Colour coordinates are based on the CIE L*a*b* system using an illuminant D65 with an observer angle of 10°. L*, a* and b*

indicate the position of the spot measured on the black/white, green/red and blue/yellow axis, respectively [34].

The ASD-FieldSpec® 3 is a compact instrumentation designed to register Visible, Near-Infrared (VNIR: 350–1000 nm) and SWIR (1000–2500 nm) reflectance spectra with a collection time of 0.1 s for each spectrum. The instrumentation is equipped with a contact reflectance probe with an internal light source, enabling the investigation of a spot area of about 1.5 cm$^2$ (Figure 4). The ASD-FieldSpec® 3 measures the reflectance of a given target by comparing it with a reference material (Spectralon, Labsphere, Inc., North Sutton, NH, USA) with known reflectance properties. In order to minimize the spectral noise, we registered 3 spectra (70 scans per spectrum) for each point, averaging them. The standard deviation of the 3 spectra on the same surface is estimated to be around $2 \times 10^{-2}$ reflectance units.

The interpretation of the SWIR spectra was based on a full profile approach, i.e., simultaneously fitting the bands and the background by an opportune model proposed by [27]. Spectra were fitted in the region between 1630 and 2500 nm. This range was chosen to avoid/minimize spectral interference at wavelength value <1630 nm [27]. In this spectral range, overtones and/or combination lines of the carbonate, sulphate and OH/H$_2$O groups of either calcite or gypsum can be easily discriminated [23–25,35]. In the fitting model, calcite and gypsum molar fraction contributions were evaluated by weighing simulated contributions of the reference compounds in a nonlinear least squares procedure [27,28]. Gypsum and calcite spectral component contributions were estimated by linear combination coefficients, $k_g$ and $k_c$, respectively, leaving unconstrained their sum. Further details on this procedural strategy can be found in [27,28].

As already operated for the colorimetric measurements, we performed a test to ascertain the statistical reproducibility of the parameterisation procedure over the repeated measurements of the two datasets. The relative uncertainty arising from this source of error was estimated in the order of magnitude of 0.004 absolute units, i.e., more than one order of magnitude below of the general uncertainty of the method [27], attributed mainly to rock texture features and to changes in the volume investigated by the SWIR radiation.

## 3. Results

### 3.1. Colorimetry

The experimental colorimetric coordinates of the twenty-four investigated spot areas of the 2012 and 2021 campaigns are reported in Table S1 (Supplementary Materials), where the average values of the L*, a* and b* parameters and their relative standard deviations (σ) are shown. In Table 1, ΔL*, Δa* and Δb* values are shown, referring to the difference between the values registered in 2021 and in 2012 for the considered parameter (e.g., $\Delta L = L_{2021} - L_{2012}$) at each point of measurement. The ΔE parameter, which is the total colour difference, was calculated as follows [36]:

$$\Delta E = \sqrt{\Delta L^{*2} + \Delta a^{*2} + \Delta b^{*2}} \tag{1}$$

to better highlight the chromatic changes on the stone surface (Table 1). The uncertainty of the resulting ΔE values (in brackets in Table 1) were obtained by error propagation theory from the uncertainty of the single parameter values (σ of the repeated measurements).

The values of ΔE clearly depict that the 2012 vs. 2021 differences for the two datasets are tracing appreciable colour changes in the considered stones. A clearer situation can be devised considering the Figure 5, where the ΔE values are plotted in a bar graph, grouped by their sampling area in the monument, as defined in Section 2 and Figure 2.

In Figure 5, the experimental ΔE values, with the relative uncertainty, are graphically compared with two lower limits, the just-noticeable differences (JND) and the uncertainty (UT) threshold levels. The JND threshold level is defined as the lowest value of ΔE above which a difference could be perceived by the human eye as "optically different". The JND value is discussed in the literature. In the field of restoration of heritage buildings, a total colour difference smaller than five units is generally considered unnoticeable to the human

eye [37–39]. However, many authors [40–42] propose smaller JND values (2–3 units range), because independent observers were reported to distinguish more subtle differences in total colour [42]. In this work, a JND value of three units was considered, which represents a very precautionary assumption.

**Table 1.** Calculated $\Delta L^*$, $\Delta a^*$, $\Delta b^*$ and $\Delta E$ (with the related uncertainty in brackets) of the 2021 versus the 2012 datasets. The data relative to the sampling point 11B are absent because of the lack of data in the 2012 database due to a failure in the saving procedure.

| Spot Areas | $\Delta L^*$ | $\Delta a^*$ | $\Delta b^*$ | $\Delta E$ |
|---|---|---|---|---|
| 1B | 1.45 | 0.74 | 2.15 | 2.7(7) |
| 2B | 3.51 | 0.18 | 0.78 | 3.6(7) |
| 3B | −6.59 | 1.71 | 4.63 | 8(1) |
| 4B | 6.36 | 0.31 | 1.38 | 7(1) |
| 5B | 0.28 | 0.32 | 2.57 | 3(2) |
| 6B | 4.29 | −0.49 | −0.23 | 4(1) |
| 7B | 9.64 | 1.83 | 2.16 | 10(3) |
| 8B | 5.97 | 1.09 | 1.88 | 6(2) |
| 9B | 11.01 | 1.02 | −3.59 | 12(1) |
| 10B | 10.83 | −1.48 | −3.63 | 12(2) |
| 11B | - | - | - | - |
| 12B | 9.80 | 0.10 | 0.86 | 10(6) |
| 13B | 0.15 | 1.06 | 3.16 | 3(3) |
| 14B | 10.51 | 0.47 | 2.41 | 11(2) |
| 15B | −3.96 | 3.52 | 6.25 | 8(3) |
| 16B | −0.96 | 1.34 | 4.24 | 5(3) |
| 17B | 4.34 | −0.24 | −0.03 | 4(2) |
| 18B | −2.39 | 1.15 | 4.96 | 6(5) |
| 19B | 4.45 | 0.27 | −0.16 | 4.5(4) |
| 20B | 7.95 | −0.16 | −0.21 | 8(1) |
| 21B | 5.14 | 0.10 | 1.72 | 5(2) |
| 22B | 4.06 | 0.36 | 0.56 | 4(2) |
| 23B | 3.10 | 1.16 | 3.16 | 5(1) |
| 24B | 2.95 | 2.26 | 3.78 | 5(1) |

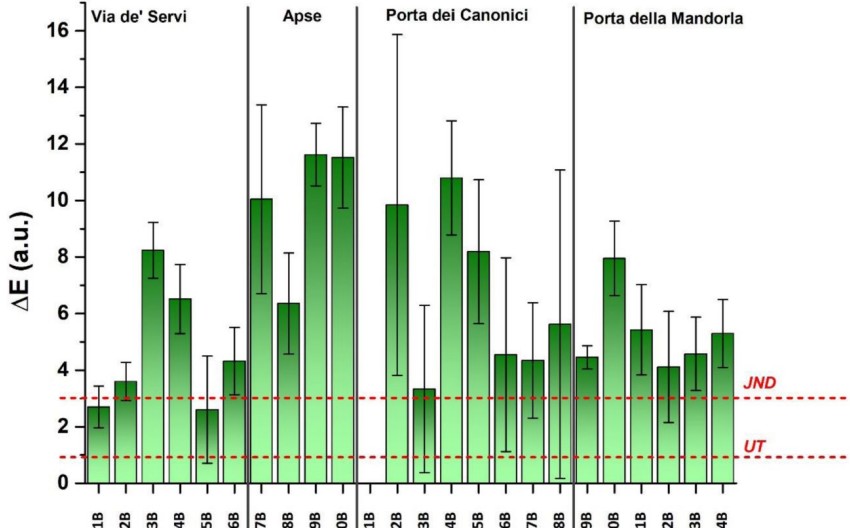

**Figure 5.** Calculated $\Delta E$ based on experimental data ($L^*$, $a^*$, $b^*$) obtained in 2012 and 2021. Vertical markers group data according to the stone localisation in the church. Experimental uncertainties are included in the graph as uncertainty bars. Horizontal red dash lines mark the JND and UT values, as defined in the text.

Conversely, the UT parameter defines the uncertainty range under which $\Delta E$ changes have no statistical meaning. We adopted for this parameter an operational definition, assuming UT defined by a $\Delta E$ shift determined by the median experimental uncertainty of the same parameter. Since the experimental distribution of the uncertainty values is not following a normal distribution, we preferred to choose the median values, in spite of the average ones.

When compared with the two reference values, almost all samples (basically all but 5B, 13B and 18B samples) fall beyond or well beyond the UT limit. The analytical procedure can sort out significant changes in the stone colour even before they are perceivable at the naked eye. Indeed, only 14 of the total 24 samples have $\Delta E$ values significantly higher than the JND level.

As depicted in Figure 6, where the three $\Delta L^*$, $\Delta a^*$ and $\Delta b^*$ parameters of Table 1 are plotted, the parameter resulting in the ability to convey the most relevant change is very often $L^*$. The results are in good agreement with the work of Grossi et al. [37], which demonstrated that the parameter $L^*$ is a good indicator to assess the soiling of stones in urban environments. In particular, $L^*$ variations were found related to the square root of the exposure time to air pollution and to the concentration of total suspended particles in the atmosphere.

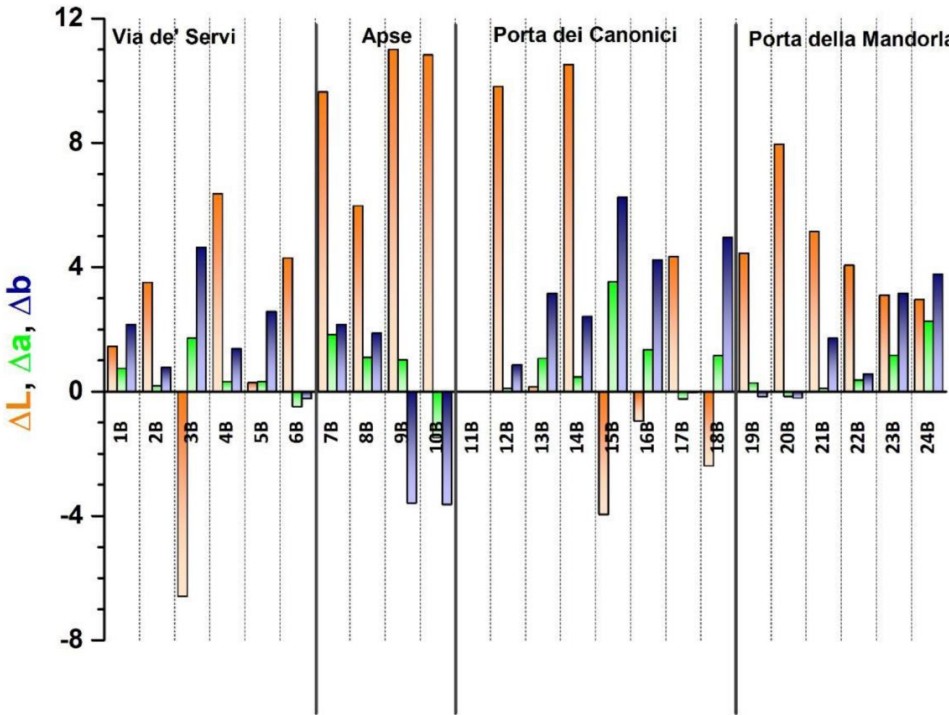

**Figure 6.** Experimental $\Delta L^*$, $\Delta a^*$ and $\Delta b^*$ (in orange, green and blue bars, respectively), as defined in the text. Vertical markers group data according to the stone localisation in the church.

Since in 2013 the Apse walls underwent an extraordinary cleaning intervention, one could tentatively relate the fact that higher $\Delta E$ and $\Delta L^*$ values are more frequently observed in samples collected from this area (7B–10B). However, a clear discrimination between the results of the Apse samples and the rest of the database cannot be fostered at present, offering the potential of further research on this field.

Samples 15B, 23B and 24B exhibit absolute $\Delta b^*$ values of the same order of magnitude or higher than the $\Delta L^*$ values. Therefore, we may conclude that the colour changes affecting the marble in the considered nine years' time span is mainly due to the shift (generally positive) of the brightness hue, while only few points exhibit a true, although small, chromatic change. The highest values of $\Delta b^*$ with respect to $\Delta a^*$ show that the colour changes much more in the component $b^*$ and, specifically, in most cases, the area

investigated has turned yellowish white. Only in the Apse area, where the cleaning has been carried out (samples 9B and 10B), does the colour turn back to cold tones.

### 3.2. SWIR Spectrometry

Selected SWIR spectra of some samples are shown in Figure 7 to exemplify the major trends observed in the spectroscopic dataset. In particular, from the comparison of the spectra registered in 2012 and 2021 on the same stone points, we can observe relevant changes in the regions of the main SWIR bands of gypsum (~1930 nm), of calcite (~2350 nm) and in the background. The two considered samples of the Figure 7a,b well depict a general observation of the whole database. The background behaviour could be severely different among different samples (compare for example that of samples in the Figure 7a,b), but it is closely reproduced in the comparison between 2012 and 2021 spectra of the same sample. Here, only a vertical shift corresponding to a wavelength independent change of reflectance in the spectra can be appreciated. A different behaviour is observed with respect to the calcite and gypsum spectral bands. Calcite contribution appears almost perfectly reproducible between 2012 and 2021, whereas for gypsum, different behaviours in different samples can be highlighted. Most samples exhibit a net or smooth decrease band intensity, as highlighted in Figure 7a,b, respectively. Lastly, in some samples, the changes in the gypsum absorption intensity do not allow a safe indication in terms of increase or decrease in gypsum contribution.

At this level of interpretation, we can generally conclude that the calcite contribution in all spectra is approximately preserved in the whole dataset, whereas gypsum contributions exhibit trends differentiated among the different samples. To have a deeper insight into these trends, the spectral parameterisation was carried out according to the procedure established by Suzuki et al. [27]. The results are listed in the Table S2 (Supplementary Materials) and graphically illustrated in S1 (Figure S1a,b). The parameterisation yielded meaningful results for all samples. According to Vettori et al. [28], a graphical representation of the gypsum over calcite molar fraction ratio could allow a fast understanding of the status of the considered samples and of the mineralogical changes undergone in the 2012–2021 time span. The gypsum/calcite ratios are shown in the Figure 8. In particular, in this plot, ratio values higher than 1 highlight spot areas where the sulfation process is in an advanced (thus critical) state, values between 1 and 0.333 spot areas where sulfation process is relevant but not alarming and ratio values below 0.333 indicate situations where sulfation, if any, is modest.

Gypsum to calcite ratios are generally lower in 2021 samples for almost all the investigated stones. As an example, the sample 3B shows ratios of 1.021 and 0.179 in 2012 and 2021, respectively. This general trend is only partially contradicted by the 2B, 4B and 5B samples, where an increase in the gypsum content is observed in 2021. However, these samples represent peculiar cases: 4B is a black region, where the full applicability of the present SWIR analysis is critical [28], while for 2B and 5B, the 2012–2021 difference, although detectable, is of the order of magnitude of the method uncertainty [27] (Figure S2). Accordingly, the dataset provides convincing evidence of a general tendency to maintain or to slightly reduce the gypsum content (thus also reducing the gypsum over calcite ratio). Considering the Apse (samples 7B–10B), where an extraordinary cleaning occurred in 2013, we can notice that a common behaviour in relation to the cleaning cannot be established: in two samples, a net decrease in gypsum content is observed (samples 7B and 8B); in another, this content is almost unchanged (sample 10B); and in the last, no gypsum was originally present since 2012 (sample 9B; cfr. Table S2 (Supplementary Materials)).

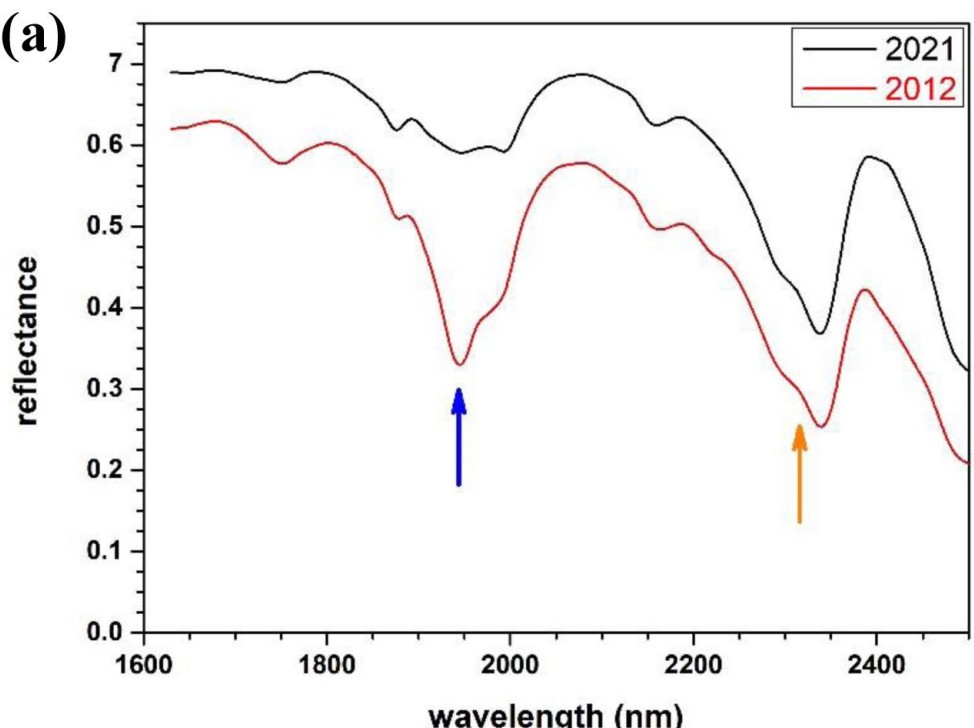

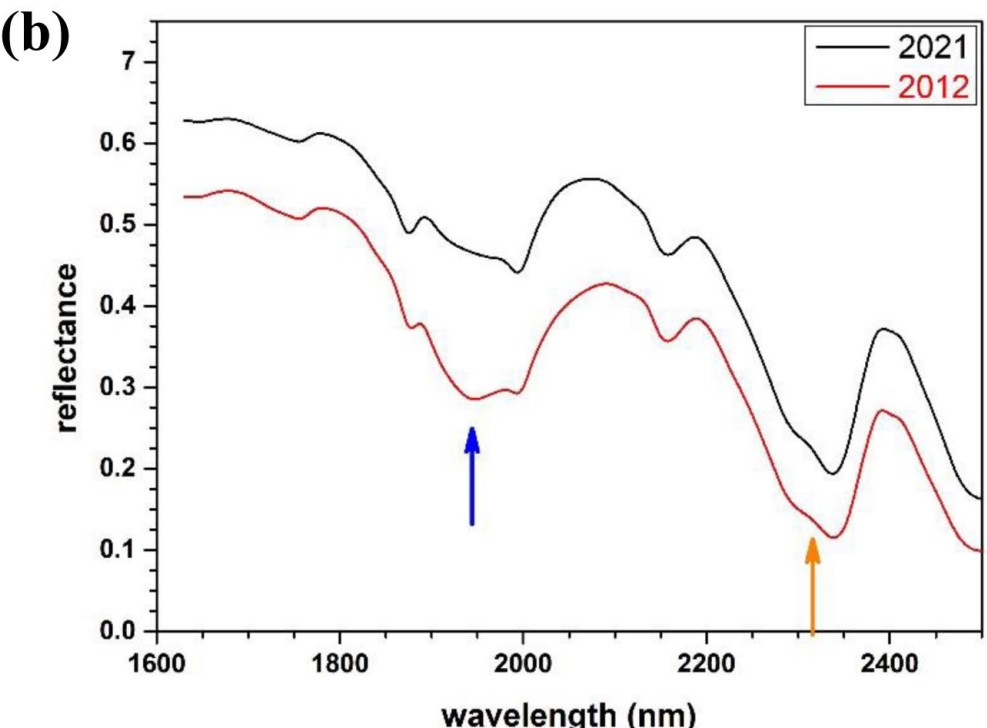

**Figure 7.** Exemplar SWIR spectra: compared 2012 (red line) and 2021 (black line) spectra for (**a**) 7B and (**b**) 12B. The blue and orange arrows mark the position of the gypsum and calcite absorption bands, respectively.

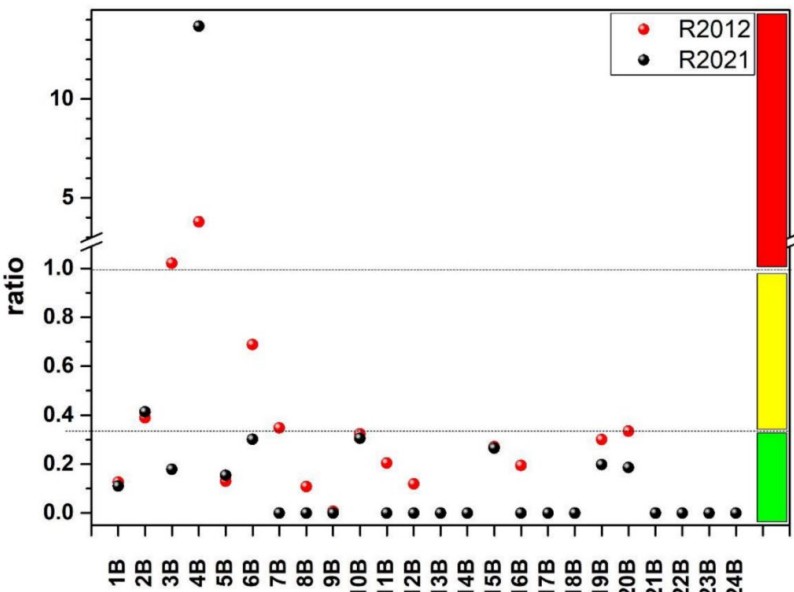

**Figure 8.** Gypsum over calcite ratio plot, including 2012 (red dots) and 2021 (black dots) data for all the analysed points. Boundaries among the different sulfation regimes, as defined in the text, are marked by the horizontal dotted lines and by the colour markers on the left side of the graph.

## 4. Discussion and Conclusions

The investigation performed in this study has pointed out that an efficient and significant combination of the proposed techniques (colorimetric and SWIR analysis) can be applied to sort out changes in the degradation process (i.e., sulfation) and to link them to the apparent changes in the colour perception of the marble stone. Indeed, the relative differences of the measurements conducted on the same areas at distance of nine years are significantly greater than the uncertainties of the analysis methods. Moreover, the protocol is rapid, repeatable and fully not invasive.

Concerning the general results obtained in this study, one can observe:

- from the colorimetric investigations, a general increase in $\Delta L^*$;
- from the SWIR investigations, a general decrease in the gypsum content.

An attempt to analyse these two phenomena as coupled did not yield meaningful results. In general, trying to interpret the occurred changes on the light of some parameters during the considered time span is beyond the purposes of the present study. However, one can provide some very qualitative discussion of the main factors which could be the main cause of the colour and mineralogical changes in the considered stones.

As already noticed, $\Delta L^*$ changes could be linked to changes on the combined effect from (a) soiling and (b) washing out of soluble/detached materials by atmospheric agents. The positive $\Delta L^*$ values, in particular, could point out that, in the case of the Santa Maria del Fiore Cathedral white marbles, where the surface porosity is negligible, the soiling particles have difficulty adhering to the surfaces, while they can be easily washed out by wind and rain. This explanation could also be supported by the closure to traffic in the cathedral area in 2009, which led to the reduction of pollution. Starting from 2009, therefore, the marble surfaces could have accumulated less soiling, while the action of atmospheric agents could have removed the degraded surface material. In the same framework, one could also give interpretation of the other changed variable: the amount of gypsum content in the stone. Such a decrease well correlates with the decrease in $S_{ox}$ $SO_x$ compounds in urban areas in the last years, and especially in Florence. This trend could have been incremented due to the decision of the civil authorities to convert the areas around the Dome into pedestrian-only streets. We are aware that the above interpretation is just hypothetical, and further investigations, based on an annual monitoring of the white marble surfaces, coupled with a mapping of all cleaning/restoration interventions, have started.

In relation to the Apse extraordinary cleaning, although a sensible colour change is recorded (the $\Delta E$ average value of the investigated area is about 8.8, quite different from that in Via de' Servi, where the $\Delta E$ average value is about 4.9), the present data seem inconclusive to identify a simple explanation process. In fact, we have uncertain results concerning the removal of the gypsum alteration (originally present in three out of four samples). In this latter case, in fact, a general trend is not observed (Figure 9). We can tentatively explain the two pieces of evidence by the fact that the cleaning procedure is tailored to remove solid particulates and biologic patinas (soiling) and not gypsum.

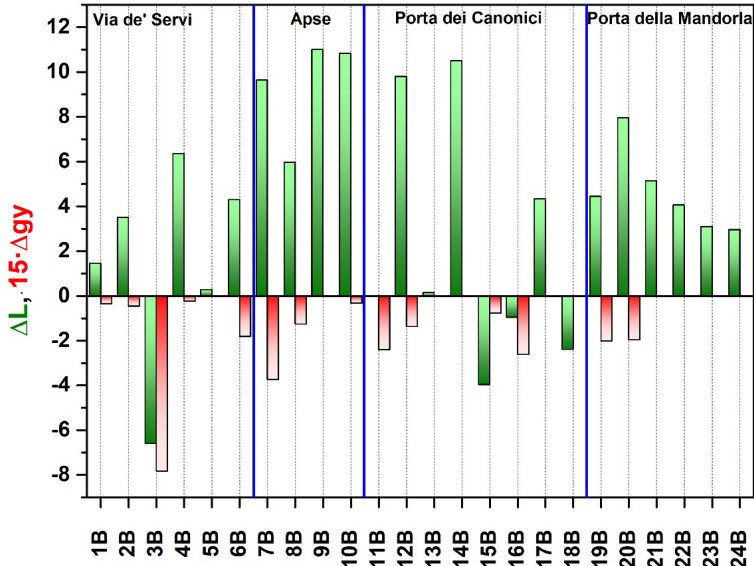

**Figure 9.** Experimental $\Delta L^*$, as defined in the text, and $\Delta gy$ (in green and red bars, respectively). $\Delta gy$ is the difference of the gypsum content between 2021 and 2012, multiplied by 15 to allow graphical comparison. Vertical markers group data according to the stone localisation in the church.

The evidence linked to this façade is in line with the qualitative interpretation provided above and it suggests that a complex mechanism dictates the evolution of the marble surface in the considered area, where probably more than one subprocess are active.

We envisage a wide applicability of the proposed protocol to establish a yearly monitoring to trace the alteration of the marble surfaces with time. To this purpose, a request to the competent authorities has been already submitted. We do envisage a wide applicability of the proposed protocol as a procedure able to provide with a limited effort and good reliability information on the evolution of the conservation state of marble cladding in monuments. This information is of paramount relevance for a conservation operated under minimal and targeted intervention in full agreement with a sustainable management of an architectural heritage.

**Supplementary Materials:** The following supporting information can be downloaded at: https:// www.mdpi.com/article/10.3390/su15021421/s1, Table S1: Mean and standard deviation (1σ, in brackets) of $L^*$, $a^*$ and $b^*$ parameters from experimental measurements in 2012 and in 2021; Table S2: Mean and standard deviation (1σ, in brackets) of calcite and gypsum weight parameters and of their ratio from experimental measurements in 2012 and in 2021; Figure S1: Results of the SWIR parameterisation: comparison of 2012 (red dots) and 2021 (black dots) data, respectively: (a) calcite molar fraction and (b) gypsum molar fraction; Figure S2: Comparison of SWIR 2012 (red line) and 2021 (black line) spectra of 5B sample. The blue and orange arrows mark the position of the gypsum and calcite absorption bands, respectively.

**Author Contributions:** Conceptualization, S.V., E.P., M.B., P.C. and F.D.B.; methodology, S.V. and F.D.B.; validation, S.V., D.R. and F.D.B.; investigation, S.V., D.R., T.S., E.P., R.M.D.F., M.C. and F.D.B.; data curation, S.V., D.R. and F.D.B.; writing—original draft preparation, S.V. and F.D.B.;

writing—review and editing, S.V., T.S., V.R., E.P., P.C. and F.D.B.; supervision, S.M., M.B. and B.A. All authors have read and agreed to the published version of the manuscript.

**Funding:** This research did not receive any specific grant from funding agencies in the public, commercial or not-for-profit sectors.

**Institutional Review Board Statement:** Not applicable.

**Informed Consent Statement:** Not applicable.

**Data Availability Statement:** Not applicable.

**Acknowledgments:** The authors thank the Opera di Santa Maria del Fiore organization for the opportunity to conduct this study. All authors have consented to the acknowledgment.

**Conflicts of Interest:** The authors declare no conflict of interest.

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
