# Peer review of "Non-Invasive SWIR Monitoring of White Marble Surface of the Cathedral of Santa Maria del Fiore (Florence, Italy)"

_sustainability, doi:10.3390/su15021421_

Round 1
Reviewer 1 Report
Likewise, I suggest to the authors, as a mere recommendation, the following modifications of the content of the work. First of all, in the introduction, briefly explain the techniques used, such as hyperspectral ones, to place the reader in the context, as well as a brief explanation of the SWIR technique used.
Second, carry out an exhaustive review of the bibliographical references to correctly adapt them to the APA standards.
Finally, it should be noted that the methodology used as well as the results obtained are very well founded.
Author Response
REVIEWER 1
Likewise, I suggest to the authors, as a mere recommendation, the following modifications of the content of the work.
First of all, in the introduction, briefly explain the techniques used, such as hyperspectral ones, to place the reader in the context, as well as a brief explanation of the SWIR technique used.
Done.
The following sentence has been added:
“Hyperspectral techniques are methods able to couple lateral resolution with energy/wavelength resolution, so as to obtain either compositional information on a region, or spectral information on a point, depending on the needs. In particular, the SWIR range is useful to highlight spectral contribution due to the intensity of bands arose from vibrational transition linked to specific mineralogical phases.”
Second, carry out an exhaustive review of the bibliographical references to correctly adapt them to the APA standards.
Done.
Finally, it should be noted that the methodology used as well as the results obtained are very well founded.

Reviewer 2 Report
The paper shows a very interesting argument, the methodology is excellent and the results are clearly described.
Author Response
REVIEWER 2
The paper shows a very interesting argument, the methodology is excellent and the results are clearly described.
The authors would like to thank the reviewer for the comments.

Reviewer 3 Report
I think it might be worthwhile to publish the article as submitted. However, I would recommend that you broaden the discussion by referring to the central themes of the journal, such as sustainability and sustainable development. Try to make connections to your case study.
Additional comments:
- References to good practices should be added in section 1.
- When reference is made to Figure 4 in the text, the specific image of the figure alluded to (a-d) should be indicated, as has been done with Figure 7.
- Are figures without authorship references supposed to be the property of the authors? This could be mentioned in the caption.
Author Response
REVIEWER 3
I think it might be worthwhile to publish the article as submitted.
However, I would recommend that you broaden the discussion by referring to the central themes of the journal, such as sustainability and sustainable development. Try to make connections to your case study.
Done.
The following sentence has been added:
“We do envisage a wide applicability of the proposed protocol, as a procedure able to provide with a limited effort, and a good reliability information on the evolution of the conservation state of marble cladding in monuments. This information is of paramount relevance for a conservation operated under minimal and targeted intervention, in full agreement with a sustainable management of an architectural heritage.”
Additional comments:
- References to good practices should be added in section 1.
As suggested by the reviewer, some references have been added to the Introduction.
- When reference is made to Figure 4 in the text, the specific image of the figure alluded to (a-d) should be indicated, as has been done with Figure 7.
Done
- Are figures without authorship references supposed to be the property of the authors? This could be mentioned in the caption.
Done
